# What Do We Know about Survival in Skeletally Premature Children Aged 0 to 10 Years with Ewing Sarcoma? A Multicenter 10-Year Follow-Up Study in 60 Patients

**DOI:** 10.3390/cancers14061456

**Published:** 2022-03-12

**Authors:** Sarah E. Bosma, Lizz van der Heijden, Luis Sierrasesúmaga, Hans J. H. M. Merks, Lianne M. Haveman, Michiel A. J. van de Sande, Mikel San-Julián

**Affiliations:** 1Department of Orthopedic Surgery, Leiden University Medical Center, 2333 ZA Leiden, The Netherlands; sarah.bosma92@gmail.com (S.E.B.); lvanderheijden@lumc.nl (L.v.d.H.); 2Department of Pediatrics, Clínica Universidad de Navarra, 31008 Pamplona, Spain; lsierra@unav.es; 3Princess Máxima Center for Pediatric Oncology, 3584 CS Utrecht, The Netherlands; j.h.m.merks@prinsesmaximacentrum.nl (H.J.H.M.M.); l.m.haveman-3@prinsesmaximacentrum.nl (L.M.H.); 4Department of Orthopedic Surgery and Traumatology, Clínica Universidad de Navarra, 31008 Pamplona, Spain; msjulian@unav.es

**Keywords:** Ewing sarcoma, pediatric, local recurrence, distant metastasis, survival, secondary malignancy, long-term outcome

## Abstract

**Simple Summary:**

Younger age has been associated with better overall survival in Ewing sarcoma, especially under the age of 10. Our study aimed at describing long-term outcomes of a cohort of 60 patients aged 0–10 with Ewing sarcoma, treated with chemotherapy, surgery and/or radiotherapy. Overall survival of these youngest patients with ES was very good. After 10 years, 81% of patients were still alive, 89% did not have a local recurrence and 81% did not have distant metastasis (in lungs and/or bone). Limb salvage surgery was achieved in >90% of patients. Wide resection margin was the only factor significantly associated with better survival, but age < 6 years, smaller tumors, no metastases at diagnosis and treatment after 2000 also seemed to result in better overall survival.

**Abstract:**

(1) Background: Younger age has been associated with better overall survival (OS) in Ewing sarcoma (ES), especially under the age of 10. The favorable survival in younger patients underlines the need for minimizing treatment burden and late sequelae. Our study aimed at describing clinical characteristics, treatment and outcome of a cohort of ES patients aged 0–10. (2) Methods: In this retrospective multicenter study, all consecutive ES patients aged 0–10, treated in four sarcoma centers in the Netherlands (*n* = 33) and one in Spain (*n* = 27) between 1982 and 2008, with a minimum follow-up of 10 years, were included. OS, local recurrence-free survival (LRFS) and distant metastasis-free survival (DMFS) were calculated. Potential factors of influence on OS (risk and protective factors) were analyzed. (3) Results: 60 patients with median follow-up 13.03 years were included. All patients were treated with chemotherapy in combination with local treatment, being surgery alone in 30 (50%) patients, radiotherapy (RT) alone in 12 (20%) patients or surgery plus RT in 18 (30%) patients (12 pre- and 6 postoperative). Limb salvage was achieved in 93% of patients. The 10-OS, -LRFS and -DMFS are 81% (95% CI: 71–91%), 89% (95% CI: 85–93%) and 81% (95% CI: 71–91%), respectively. Six patients developed LR, of which two developed subsequent DM; all had axial ES (pelvis, spine or chest wall), and these patients all died. Ten patients developed DM; eight died due to progressive disease, and two are currently in remission, both with pulmonary metastasis only. Negative or wide resection margin was significantly associated with better OS. Age < 6 years, tumor volume < 200 mL, absence of metastatic disease and treatment after 2000 showed trends towards better OS. Two patients developed secondary malignancy; both had chemotherapy combined with definitive RT for local treatment. (4) Conclusions: Overall survival of these youngest patients with ES was very good. Limb salvage surgery was achieved in >90% of patients. Wide resection margin was the only factor significantly associated with better survival.

## 1. Introduction

Ewing sarcoma (ES) is an aggressive bone and soft-tissue tumor predominantly affecting children and young adults [1,2,3,4]. Fifteen to 20% of patients are aged between 0 and 10 years at time of diagnosis [5,6,7]. Advances in multimodal treatment of Ewing sarcoma have gradually improved survival to a 10-year overall survival (OS) of 70–75% in nonmetastatic Ewing sarcoma. Survival in metastatic Ewing sarcoma remains poor, with a 5-year OS of 20–35% [8,9,10].

Known risk factors for worse outcome in Ewing sarcoma are primary tumors located in the pelvis or spine [11,12,13,14,15] and chest wall [16,17], as well as large tumor volume (>200 mL) and size (>8 cm) [18,19,20,21] and presence of metastatic disease [3,22,23]. Patients with extrapulmonary metastases do significantly worse than patients with pulmonary metastases alone [8,19,24]. Contrarily, in multivariate analyses, younger age has been associated with a better clinical outcome, especially under the age of 10 [12,19,22,24,25].

It remains unclear why children under the age of 10 with Ewing sarcoma would have a better survival. A large study showed that young children (0 to 9 years old) are less likely to present with primary tumors of the pelvis or spine and less often have metastatic disease at diagnosis [6]. However, whether age has prognostic value independently from its association with other variables predicting poor outcome, such as metastatic disease and tumor site, is still unclear. However, the favorable survival in patients under the age of 10 underlines the need for minimizing long-term morbidity and investigating treatment burden. Individual ES treatment should be decided on based on overall estimated survival, burden of therapy experienced and predicted late sequelae.

The purpose of this retrospective multicenter study was to describe clinical characteristics, treatment and outcomes of a cohort of Ewing sarcoma patients aged 0 to 10 years with a minimum follow-up of 10 years.

## 2. Materials and Methods

### 2.1. Study Design and Patients

This retrospective study was approved by the local ethical board and granted a waiver for the requirement of informed consent. The national database of the Dutch Childhood Oncology Group (DCOG) and the local database of Universidad de Navarra (Pamplona, Spain) were searched for patients diagnosed with Ewing sarcoma between 1982 and 2008. Patients were eligible for inclusion when meeting the following criteria: (1) histopathological confirmed Ewing sarcoma; (2) aged 0 to 10 years at time of diagnosis; (3) minimum follow-up of 10 years or death within 10 years.

Patient characteristics included age at diagnosis, gender, primary tumor localization, disease extent, tumor size and volume. Axial tumor location includes tumors located in the pelvis, spine and chest wall. Volume was calculated according to the formula diameter × height × depth × 0.52 (LR × AP × CC × 0.52) [26,27]. Treatment characteristics included chemotherapy protocol, local treatment modality, radiotherapy dose and timing, type of surgery, surgical margins, histological response (percentage of necrosis) and follow-up data on local recurrence, distant metastasis, incidence of secondary malignancies and death.

Histological response (percentage necrosis) and resection margins were assessed on the surgical specimen by experienced local pathologists. Surgical margins were classified as Free (R0) if the tumor was completely removed during surgery, not damaged and covered by intact lining of normal tissue or “capsule” both macro- and microscopically. Margins were considered marginal (R1) in case the tumor was macroscopically completely removed and not damaged during surgery, but microscopically, tissue reached resection margins without clear evidence of residual tumor in situ. Margins were classified as intralesional (R2) in case of incomplete removal or damage of the tumor during surgery or if the tumor tissue reached the resection margin with evidence of residual tumor in situ. The degree of histological response was defined by the percentage of viable tumor cells in the specimen. Data were analyzed using three groups: (1) 100% necrosis; (2) 90–99% necrosis; (3) <90% necrosis. Follow-up in terms of local recurrence, distant metastasis and OS was calculated from the date of diagnosis. Local recurrence was defined as local-regional recurrence after initial complete response. Distant metastasis was defined as new metastatic disease or recurrence of metastatic disease after initial complete response.

### 2.2. Statistics

Patient and tumor characteristics were summarized using descriptive statistics. Overall survival (OS), local recurrence-free survival (LRFS), distant metastasis-free survival (DMFS) and event-free survival (EFS) were measured from initial date of diagnosis until last day of follow-up or date of death and analyzed using Kaplan–Meier statistics and log-rank test. Potential factors of influence on OS, including risk factors and protective factors, were assessed using Cox regression univariable analysis. IBM Statistical Package for Social Statistics 25 (SPSS, Chicago, IL, USA) was used for analysis.

## 3. Results

A total of 60 patients (27 from Spain, 33 from the Netherlands) were included (Table 1). Median follow-up, assessed by the reverse Kaplan–Meier method, was 13.03 years (95% confidence interval (CI): 11.02–15.05 years). All patients were treated with a combination of chemotherapy and local tumor treatment, being either surgery, radiotherapy or both. There was a standard difference in approach between the Netherlands and Spain.

In the Netherlands, patients were treated according to the Cooperative Ewing Sarcoma Study Group from 1991 to 2009 (CESS-86, EICESS-92 and EURO-E.W.I.N.G.99). Twenty-eight patients (47%) were treated according to the EURO-E.W.I.N.G.99 protocol, with six cycles of vincristine, ifosfamide, doxorubicin, etoposide (VIDE) induction chemotherapy followed by local treatment of the primary tumor. After local treatment, patients received maintenance therapy, consisting of one cycle of vincristine, actinomycin D and ifosfamide (VAI) followed by either seven cycles of VAC (cyclophosphamide instead of ifosfamide) or VAI for standard-risk patients (nonmetastatic and <200 mL initial tumor volume). High-risk patients (nonmetastatic and >200 mL initial tumor volume or patients with pulmonary metastasis only) received seven cycles of VAI or high-dose treatment with busulfan and melphalan followed by autologous stem cell transplantation (SCT). Three patients (5%) were treated according to the EICESS-92 protocol. In EICESS-92, standard-risk patients (nonmetastatic and <100 mL initial tumor volume) had four 3-week courses of vincristine, doxorubicin, ifosfamide, actinomycin D (VAIA) and were then randomized to receive another 10 courses of either VACA (cyclophosphamide instead of ifosfamide) or VAIA. High-risk patients (metastatic or >100 mL initial tumor volume) were randomized to receive either VAIA or VAIA plus etoposide (EVAIA). Two (3%) patients were treated according to the CESS-86 protocol. In CESS-86, patients classified as standard-risk with nonmetastatic tumors of the extremity and <100 mL initial volume received VACA; all other patients received VAIA.

In Spain, patients were treated according to the local chemotherapy treatment protocols. Seventeen patients (28%) received seven cycles of vincristine, doxorubicin and cyclophosphamide (VDC), alternating with seven cycles of VC + bleomycin. Ten patients (17%) received seven cycles of VDC, alternating with seven cycles of VIE.

### 3.1. Surgery

Overall, 48 (80%) patients underwent surgery of the primary tumor. Limb salvage was achieved in 93% of the cases. Of the 12 patients that did not get surgery, 9 had tumors located in the axial skeleton (7 in pelvis, 1 in spine, 1 rib); 4 presented with metastasis disease at diagnoses, of which 3 presented with bone metastasis and one with combined pulmonary and bone metastasis.

Nineteen patients (32%) underwent resection without reconstruction. Thirteen of these tumors were located in the axial skeleton, 4 in the soft tissue only and 2 in the extremity (ulna and fibula). Tumor volume was <200 mL in 13 patients, and tumor size was <8 cm in 10 patients.

Fourteen patients (23%) had limb salvage surgery with allograft reconstruction. All allografts are still in situ. Six patients had an epiphysiolysis before graft reconstruction. Three out of 14 patients had fractures of the allograft but no problems after refixation. One patient had an 8 cm leg length discrepancy that was corrected by shortening of the contralateral leg. One patient developed a genu valgum that was corrected. The other nine allografts showed no problems 10 years or more after treatment.

Six (10%) patients underwent reconstruction with an autograft; all these tumors were located in the lower extremity. Five (8%) patients underwent resection followed by reconstruction with endoprosthesis; all these tumors were located in the lower extremity. Only two patients underwent amputation, one of the forefoot and one below-knee amputation. Last, two patients underwent rotationplasty (Van Nes Borggreve).

### 3.2. Radiotherapy

Overall, 32 (53%) patients were treated with radiotherapy. Eighteen (30%) patients received chemotherapy in combination with surgery and radiotherapy, of which 12 (20%) received preoperative chemotherapy and 6 (10%) postoperative. Patients that had received surgery in combination with radiotherapy generally had large tumor volumes (11 ≥ 200 mL versus 7 < 200 mL) and large tumor size (14 ≥ 8 cm versus 4 < 8 cm). Ten of these tumors were located in the extremities, 7 in the axial skeletal (pelvis, spine or chest wall) and 1 in the soft tissue only. Margin status after surgery was wide in 14 patients and marginal in 4. Histological response was 100% necrosis in 11 patients, 90–99% necrosis in 3 patients, <90% necrosis in 1 patient.

Twelve (20%) patients received definitive radiotherapy. Seven of these tumors were located in the axial skeleton (five pelvic; one spine; one scapula), two in the extremities, two in the soft tissue (one retroperitoneal and one soft tissue of the upper extremity) and one cranial. Volume was ≥200 mL in nine patients.

Two patients (3%) underwent surgical resection followed by high-dose (80 Gy and 120 Gy) extracorporeal radiotherapy of the resection specimen and subsequent reconstruction with the irradiated autograft bone. These tumors were located in the calcaneus and scapula.

### 3.3. Survival, Local Control and Distant Metastases

Fourteen patients (23%) died, 13 due to progressive disease and 1 due to secondary malignancy (Table 1 and Table 2). The 5-, 10- and 15-year OS is 83% (95% CI: 78–88%), 81% (95% CI: 71–91%) and 75% (95% CI: 63–87%), respectively.

The 5-, 10- and 15-year LRFS is 93% (95% CI: 87–99%), 89% (95% CI: 85–93%) and 89% (95% CI: 85–93%), respectively. There was no difference in OS between countries with different treatment protocols (10-year OS: the Netherlands, 82% (95% CI: 69–95); Spain, 81% (95% CI: 66–96); log-rank *p* = 0.342).

Six patients (10%) developed local recurrence, of which two developed subsequent distant metastasis (one pulmonary metastasis only and one combined bone and pulmonary metastasis). All these six patients died due to progressive disease. Location of their primary tumor was chest wall in three cases, pelvic in two cases and spine in one case. Two of these patients presented with metastatic disease at diagnosis, one with pulmonary metastasis and one with combined pulmonary and bone metastasis. Three patients were treated with definitive radiotherapy; one received postoperative radiotherapy. Three patients were treated with surgery alone, resection margins were wide and histological response was 100% in one patient and 90–99% in the other two patients.

Ten patients (17%) developed distant metastasis after finishing treatment. Two of these 10 patients already had bone metastasis at diagnosis and developed new metastasis during the course of treatment. Three patients developed pulmonary metastasis and seven combined (pulmonary with bone or other). Two of those patients are currently in remission, both with pulmonary metastasis only. The other eight died due to progressive disease. The 5-, 10- and 15-year distant metastasis-free survival (DMFS) is 83% (95% CI: 73–93%), 81% (95% CI: 71–91%) and 79% (95% CI: 68–89%), respectively.

### 3.4. Secondary Malignancy

Two out of 60 patients developed a secondary malignancy; both of these patients had chemotherapy combined with definitive RT (of which 1 proton beam) as local treatment. One patient developed myelodysplastic syndrome-refractory anemia with excess blasts (MDS-REAEB) 6 years after initial diagnosis and treatment with etoposide. The patient was treated with chemotherapy and definitive radiotherapy with a total dose of 62 Gy. The patient is currently in remission at a follow-up of 14 years. The other patient developed a high-grade undifferentiated pleomorphic soft-tissue sarcoma in the radiation field 13 years after initial diagnosis. The tumor was treated with definitive radiotherapy with a total dose of 55.8 Gy. Three years after diagnosis of the secondary malignancy, the patient died due to progressive disease.

### 3.5. Prognostic Factors

Free (R0) resection margin was the only factor significantly associated with better OS in univariate risk analysis (Table 3). Age below 6 years, tumor volume < 200 mL, absence of metastatic disease at diagnosis and treatment era after 2000 showed a trend towards better OS (Figure 1).

## 4. Discussion

Overall survival in Ewing sarcoma seems worse in older adolescents (15–19 years) compared with the youngest children (0–10 years) [6,28,29]. In addition, 40% of late mortality in ES (>5 years after treatment) is due to secondary malignancies, cardiac disease or renal failure [30]. Radiotherapy is associated with a significant risk of secondary radiotherapy-induced malignancies (lifetime risk varying from 2 to 11%). Even in the case of low-dose whole lung irradiation, the risk of breast cancer is elevated. Surgery of ES, on the other hand, comes with long-term functional problems [28,30,31]. Thus, multimodality ES therapy is associated with significant acute toxicities and long-term effects. Especially in the youngest children, who have a better OS, this presents a dilemma as to what the best approach is to optimize chance of cure, minimize toxicity and respect quality of life. To gain more insight into factors determining favorable outcome in younger children with ES and in long-term treatment and resulting adverse effects because of their favorable survival, the purpose of this retrospective multicenter study was to describe clinical characteristics, treatment and outcomes of a cohort of Ewing sarcoma patients aged 0 to 10 with a minimum follow-up of 10 years.

A large study including data from 2635 patients showed that young children (0 to 9 years old, *n* = 563) present with a lower proportion of pelvic and axial primary tumors and less often with metastatic disease at diagnosis [6]. To date, biological and developmental differences between age groups in ES have not been clarified, and it remains unknown why less axial and pelvic ES are seen in younger children. Better outcome in the young patients could be explained by higher relative doses of chemotherapy (mg/m^2^) [32] or radiotherapy (Gy) in younger children; better local tumor control may be achieved, but at the cost of higher long-term morbidity, including the risk of secondary malignancy. Therefore, the balance between survival and the toxicity of intensive salvage treatments could be (re)considered.

It can also be hypothesized that parents monitor their youngest children closer and seek medical assistance at an earlier stage in case of withholding the use of an arm or leg, limping or soft tissue swelling, resulting in less patient (or parent) delay. Maybe in younger adolescents (10–14 years) and older adolescents (15–19 years), concomitant sports injuries or surmenage may overshadow or mimic complaints of underlying malignancy or complaints are reported at a later stage. This might result in longer patient delay and thus larger tumor size and more frequent metastases at time of diagnosis. In pediatric osteosarcoma and ES, time from initial symptoms to start of treatment matters in terms of survival and local recurrence rates [33,34]. However, in both osteosarcoma and ES, it has also been suggested that treatment delay would not influence outcome, so maybe biological behavior of these malignancies is more important for prognoses [35,36].

In our study, patients had better OS, LRFS and DMFS compared to older ES patients in the literature. We report 5-, 10- and 15- year OS of 83%, 81% and 75%; LRFS of 93%, 89% and 89%; and DMFS of 83%, 81% and 79%, respectively. Surgical margins were the only strong risk factor associated with worse OS, with a 13-fold increased risk in patients without R0 resections. A trend was seen towards improved survival for the youngest children aged 0–6 years when compared with aged 6 to 10 years. There are only two other studies on survival of younger children with ES in specific, albeit with short median follow-up and without late treatment effects, including growth deficiencies and functional limitations. De Ioris et al. reported 62 ES patients aged 0–6 (1990–2008) [37]. Huh et al. reported 42 ES patients aged 0–10 (1980–2010) [38]. Precisely 34–41% had extremity localizations, and 17–23% had metastases at diagnosis. Median follow-up was 4.7–5.2 years (1 month–30 years). Five-year OS and LRFS were 73–82% and 67–72% for nonmetastatic ES and 38% and 21% for metastatic ES. Remarkably, patients included in the final decade of De Ioris’ study (i.e., after 2000) had a better 5-year OS of 89% and PFS of 86%, in accordance with more recent literature. Huh et al. investigated different age groups, but reported no differences in OS and RFS between children aged 0–5 and 6–10 years.

Huh et al. found that metastasis at presentation was the only risk factor for decreased OS (43% versus 88%) and RFS (29% versus 73%) [38]. De Ioris et al. also reported worse survival for metastatic disease [37]. In our study, presence of metastases at diagnosis did not significantly worsen outcome, possibly because of lower incidence in these youngest children (only 18% had metastatic disease at diagnosis of which 64% pulmonary metastasis only) and limited cohort size.

In our series, nonirradiated patients had no worse OS. Usually, radiotherapy in the youngest children is avoided in order to diminish its secondary effects on length growth and the possibilities of provoking radiation-induced sarcoma (3–12%) [39,40,41]. Adjuvant radiotherapy reduces local recurrence rate, especially in larger tumors (>200 mL or >8 cm in this study), poor histological response or inadequate surgical margins [42]. Definitive radiotherapy was mostly indicated for axially (pelvis, spine or chest wall) or retroperitoneally located ES in our study and should be reserved in case complete surgical excision is impossible. High-dose extracorporeal radiotherapy on the resection specimen and subsequent reconstruction with the irradiated autograft may be performed [43,44], although not available in every sarcoma center.

The majority of patients in our series underwent surgical resection of their primary ES, and limb salvage was achieved in 93%. In specific tumor localizations, reconstruction may not be required (e.g., proximal fibula, distal ulna, iliac wing, soft tissue); this was the case in one-third of our patients. There are several options at hand when reconstruction is indicated, including massive allograft with or without epiphysiolysis beforehand, free vascularized fibula autograft and (growing) endoprosthesis, all with their accompanying risks and benefits [45]. Most allografts are in situ at long-term follow-up. Leg length discrepancies are common in all reconstruction types due to the large remaining growth potential in the youngest children, often requiring multiple surgeries of the affected (or contralateral) limb. Finally, in our series including the youngest patients, amputation or rotationplasty was performed in only 7% of patients.

Secondary malignancies may arise in up to 2–12% of all patients treated for childhood sarcoma [46,47,48,49,50]. We reported 2 out of 60 patients with secondary malignancy (3%) after a minimum follow-up of 10 years. Huh et al. reported 2 out of 62 patients with secondary malignancy (3%), with a median follow-up of 5 years and the shortest follow-up of 8 months, potentially resulting in an underestimation.

### Limitations

Even if, to the authors’ best knowledge, this is the largest patient series in children aged 0–10 years with ES reporting long-term follow-up, it remains a small cohort with a retrospective design and its forthcoming implications. For example, no valid (subgroup) analysis could be performed of various local treatment strategies used, and due to the small number of events, no multivariable risk analysis could be performed. However, we feel that the present study provides useful information on a rare subgroup of patients with a long minimum follow-up of 10 years.

## 5. Conclusions

We described clinical characteristics, treatment and outcomes of a cohort of 60 Ewing sarcoma patients aged 0–10 with a minimum follow-up of 10 years. Overall survival of these youngest patients with ES was actually very good and was estimated at 83%, 81% and 75% after 5, 10 and 15 years, respectively. Limb salvage surgery was achieved in >90% of patients. Free resection margin was the only factor significantly associated with better overall survival. Age < 6 years, tumor volume < 200 mL, absence of metastasis at diagnosis and treatment after 2000 showed trends towards better survival. Prognosis of ES indeed seems better in the youngest children, and even without radiotherapy, prognosis remains good. In this specific group, and especially in the very young, addition of radiotherapy should be well balanced between potential benefit in survival and expected late adverse effects.

## Figures and Tables

**Figure 1 cancers-14-01456-f001:**
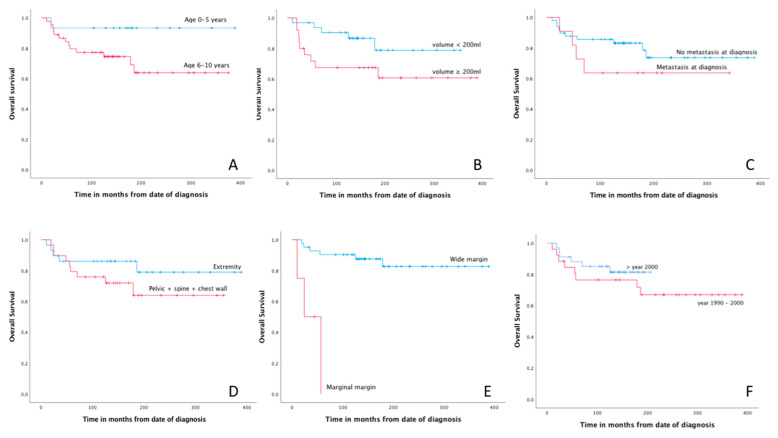
Kaplan–Meier curves for age (**A**), tumor volume (**B**), metastasis at diagnosis (**C**), tumor location (**D**), surgical margin (**E**) and treatment era (**F**).

**Table 1 cancers-14-01456-t001:** Patient demographics of all patients and a subset of patients that died.

	All Patients	Deceased Patients
	N (%)	N (%)
**Total**	60	14 (23)
**Gender**		
Male	37 (62)	6 (43)
Female	23 (38)	8 (57)
**Age (year) (mean, SD)**	7 (2.7)	8.4 (1.9)
**Location primary tumor**		
Extremity	29 (48)	5 (36)
Upper extremity	5 (8)	3 (22)
Lower extremity	24 (40)	2 (14)
Axial:	22 (37)	9 (63)
Pelvic	10 (17)	3 (22)
Spine	1 (2)	1 (7)
Chest wall (costa/sternum/scapula)	11 (18)	5 (36)
Other		
Retroperitoneal, soft tissue	6 (10)	
Skull/cranial	3 (5)	
**Volume**		
<200 mL	32 (53)	5 (36)
≥200 mL	25 (42)	9 (63)
Missing	3 (5)	
**Size**		
<8 cm	21 (35)	3 (21)
≥8 cm	36 (60)	11 (79)
Missing	3 (5)	
**Metastasis at diagnosis**		
No	49 (82)	10 (71)
Lung metastasis	8 (13)	1 (7)
Bone metastasis	3 (5)	3 (21)
**Treatment**		
Radiotherapy	32 (53)	11 (79)
Definitive	12 (20)	6 (43)
Preoperative	12 (20)	4 (29)
Postoperative	6 (10)	1 (7)
Extracorporeal RT	2 (3)	
Dose (Gy) (mean, SD)	49 (17.9)	
Surgery	48 (80)	8 (57)
Rotationplasty	2 (3)	1 (7)
Allograft	14 (23)	2 (14)
Autograft	6 (10)	
Prosthesis	5 (8)	1 (7)
Amputation	2 (3)	
Resection without reconstruction	19 (32)	4 (29)
**Surgical margin**		
R0	42 (70)	5 (36)
R1	4 (7)	3 (21)
Other *	2 (3)	
**Histological response**		
100%	27 (45)	5 (36)
90–99%	13 (22)	2 (14)
<90%	2 (3)	1 (7)
Missing	6 (10)	

Abbreviations: N = number of patients; SD = standard deviation. Continuous variables are presented by the mean with corresponding standard deviation between brackets, categorical variables as a number with the percentage between brackets. *: extracorporeal RT and reimplantation.

**Table 2 cancers-14-01456-t002:** Number of deceased patients in each treatment group.

	Surgery Alone	Radiotherapy Alone	Surgery Combined with Radiotherapy
All patients	30	12	18
Deceased patients	3 (10%)	6 (50%)	5 (28%)

**Table 3 cancers-14-01456-t003:** Results of univariate Cox regression analysis at time of diagnosis ^1^ and time of surgery ^2^.

Variables	Univariate Analysis	
	HR (95% CI)	*p*-Value
**Gender ^1^**		
Male	1	
Female	2.51 (0.87–7.24)	0.088
**Age ^1^**		
0–5 years	1	
6–10 years	5.06 (0.66–38.69)	0.119
**Volume ^1^**		
<200 mL	1	
≥200 mL	2.55 (0.85–7.64)	0.095
**Location ^1^**		
Extremity	1	
Axial (spine + chest wall)	1.84 (0.44–7.75)	0.404
Pelvic	1.88 (0.57–6.17)	0.300
**Metastasis at diagnosis ^1^**		
No	1	
Yes	1.84 (0.58–5.86)	0.306
**Histological response ^2^**		
100%	1	
90–99%	0.90 (0.18–4.65)	0.901
<90%	2.15 (0.25–18.65)	0.489
**Surgical margin ^2^**		
R0	1	
R1	13.57 (2.92–63.04)	**<0.001**
**Treatment era**		
>2000	1	
1990–2000	1.56 (0.53–4.60)	0.423

Abbreviations: HR = hazard ratio; CI = confidence interval.

## Data Availability

The data presented in this study are available on request from the corresponding author.

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
