# Peer review of "What Do We Know about Survival in Skeletally Premature Children Aged 0 to 10 Years with Ewing Sarcoma? A Multicenter 10-Year Follow-Up Study in 60 Patients"

_cancers, 2022, doi:10.3390/cancers14061456_

Round 1

Reviewer 1 Report

Excellent review of 60 patients with Ewing sarcoma under the age of 10. Please review the following: Line 32: Surgery plus RT was done in 18 and not 32 patients as currently written here. Authors mentioned 18 patients with disease recurrence then it was stated that 10 patients developed distant metastasis and 6 developed LR. Can you please explain this discrepancy? In the table: location: 25 axial, 29 extremities, skull/cranial 3, retroperitoneal 6. This totals 63, I presume due to cases presenting with muticentric disease. Can you please make this clear in your manuscript? Can you please explain why, in your opinion, patients after 2000 did better? Line 240: please change "insight in" to "insight into" Line 325: please delete extra-space after "ES"

Author Response

Dear reviewer,

Thanks for your positive remarks and sharp analytics, also with the numbers. We changed all your suggestions accordingly with track changes in the text, and in the document you can find our specific answers.

Reviewer 2 Report

This is a nice study describing the clinical characteristics and outcomes of young Ewing sarcoma patients which is relevant although multivariable risk analysis could not be performed due to the small number of events.

Comments:

  • In the abstract (line 32) you state that 32 patients were treated with surgery plus RT. This number is not correct since the total number of patients in that case exceeds 60. This number has to be 18 according to the information in the article. The % is also incorrect.
  • Line 36 of the abstract states "all patients died". I would rephrase this into "these patients all died".
  • Throughout the article you start many sentences with a number. Try to avoid that by rephrasing or by writing the number in text.
  • Line 106: RFS is used here whereas LRFS is used throughout the article.
  • Line 107: LRFS and DMFS were measured from date of diagnosis. Do you mean the initial diagnosis or diagnosis of LR or DM?

Author Response

(The authors gave the same response as above.)
